# MangaVQA and MangaLMM:
# A Benchmark and Specialized Model for Multimodal Manga Understanding

**Jeonghun Baek**[*]  **Kazuki Egashira**[*]  **Shota Onohara**[*]  **Atsuyuki Miyai**[*]
**Yuki Imajuku**    **Hikaru Ikuta**    **Kiyoharu Aizawa**
The University of Tokyo

https://github.com/manga109/MangaLMM/

## Abstract

Manga, or Japanese comics, is a richly multimodal narrative form that blends images and text in complex ways. Teaching large multimodal models (LMMs) to understand such narratives at a human-like level could help manga creators reflect on and refine their stories. To this end, we introduce two benchmarks for multimodal manga understanding: MangaOCR, which targets in-page text recognition, and MangaVQA, a novel benchmark designed to evaluate contextual understanding through visual question answering. MangaVQA consists of 526 high-quality, manually constructed question–answer pairs, enabling reliable evaluation across diverse narrative and visual scenarios. Building on these benchmarks, we develop MangaLMM, a manga-specialized model finetuned from the open-source LMM Qwen2.5-VL to jointly handle both tasks. Through extensive experiments, including comparisons with proprietary models such as GPT-4o and Gemini 2.5, we assess how well LMMs understand manga. Our benchmark and model provide a comprehensive foundation for evaluating and advancing LMMs in the richly narrative domain of manga.

## 1  Introduction

Manga is a rich and distinctive form of multimodal narrative, combining complex panel layouts, expressive visual elements, and text embedded directly within images. As large multimodal models (LMMs) continue to advance in vision-language understanding, enabling them to understand manga presents an exciting opportunity, not only as a technical milestone, but also as a way to support human creativity. Such models could assist manga creators in reflecting on and refining their stories. To provide meaningful assistance, an LMM would need to function like a skilled editor or assistant, capable of reading and understanding manga in a way human does. This calls for evaluating models' abilities to process visual-textual content and follow the context in a coherent and human-like manner.

Although recent efforts such as Magi [25, 24, 26] and CoMix [30] have tackled comic understanding, they primarily focus on generating transcriptions from comic pages – they do not evaluate to what extent models can accurately read in-page text using optical character recognition (OCR), or understand the content based on that text through visual question answering (VQA). As a result, it remains unclear to what extent models truly comprehend manga content in a human-like manner based on the embedded textual information.

To pave a reliable path toward comprehensive manga understanding in LMMs, we believe it is essential to evaluate two core capabilities: OCR and VQA. To address these needs, we propose

---

[*]Equal contribution.

Submitted to 39th Conference on Neural Information Processing Systems (NeurIPS 2025). Do not distribute.

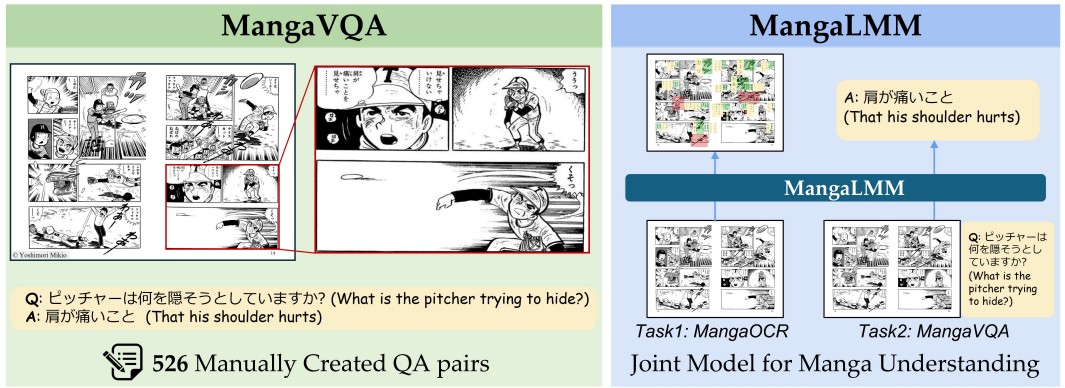

Figure 1: **Overview of MangaVQA and MangaLMM.** We present MangaVQA, a newly proposed benchmark for multimodal context understanding, consisting of 526 manually constructed question–answer pairs. We also develop MangaLMM, a manga-specialized model jointly trained to handle both MangaOCR and MangaVQA tasks.

two benchmarks: MangaOCR and MangaVQA. **MangaOCR** focuses on detecting and recognizing textual content such as dialogue and sound effects. We consolidate existing annotations from the well-known Manga109 dataset [20, 2] and the manga onomatopoeia dataset [3] to construct this benchmark. Further, as our primary contribution, we propose **MangaVQA**, a novel benchmark designed to evaluate an LMM's ability to accurately answer targeted, factual questions grounded in both visual and textual context. It consists of 526 high-quality, manually constructed question–answer pairs covering a diverse range of scenarios, enabling assessment of a model's narrative understanding. Together, these benchmarks provide a comprehensive framework for evaluating a model's ability to understand manga as a multimodal narrative medium, with MangaVQA playing a central role in assessing deeper semantic and contextual comprehension.

Furthermore, truly human-like understanding of manga requires the ability to jointly perform both OCR and VQA, rather than treating them as isolated tasks. Therefore, building on our two proposed benchmarks, we finetune an open-source LMM (Qwen2.5-VL [4]) to develop **MangaLMM**, a manga-specialized model designed to jointly address both OCR and VQA tasks. MangaLMM serves as a practical baseline for human-like manga understanding. We conduct comprehensive experiments, including analyses on model and dataset size, and compare MangaLMM with state-of-the-art proprietary models such as GPT-4o [12] and Gemini 2.5 [9] to evaluate the current landscape of multimodal manga understanding. Our results show that even the proprietary models struggle on our two benchmarks, while MangaLMM jointly handle OCR and VQA, achieving promising performance on both.

An overview of our proposed MangaVQA benchmark and the MangaLMM model is shown in Figure 1. Our contributions are summarized as follows:

- We present **MangaVQA**, a novel benchmark for evaluating multimodal question answering in manga, consisting of 526 manually constructed question–answer pairs. Combined with **MangaOCR**, which focuses on precise, in-page text detection and recognition—an aspect often overlooked in prior comic-related benchmarks, our benchmarks provide a foundational evaluation of multimodal manga understanding across both visual and textual dimensions.
- We develop **MangaLMM**, a manga-specialized version of Qwen2.5-VL finetuned on synthetic VQA and MangaOCR annotation, designed to jointly address both VQA and OCR.
- We perform extensive analysis on how model size and training data influence performance, and evaluate MangaLMM against proprietary models such as GPT-4o and Gemini 2.5 to assess the limitations of general-purpose LMMs in stylized visual domains.

## 2  Related Work: Comic Datasets and Tasks

Recent work, CoMix [30], has unified various comic-related tasks by analyzing existing datasets, including French comics (eBDtheque [10]), American comics (COMICS [14] and DCM772 [23]),

and Japanese comics (Manga109 [20] and PopManga [25]). CoMix primarily focuses on transcript generation-related tasks, including object detection, speaker identification, character re-identification, reading order prediction, and character naming prediction. Similarly, the recent Magi series (v1 [25], v2 [24], and v3 [26]) also centers on transcript generation. Notably, Magi v3 extends this pipeline by generating image captions from transcriptions and further producing prose based on those captions.

Although recent studies such as CoMix and the Magi series have addressed a wide range of tasks, the evaluation of OCR has often been underexplored, particularly in detecting the locations of texts within an image and recognizing their content. One exception is COMICS TEXT+ [28], which evaluates OCR performance at the panel level, but it does not address page-level evaluation. However, humans typically perceive and interpret text at the page level, integrating visual and textual cues across the entire layout. To reflect this human reading process, we evaluate OCR performance on two-page spreads using MangaOCR.

Existing studies have also largely overlooked the visual question answering (VQA) task in the context of comics. Among prior datasets, the Manga Understanding Benchmark (MangaUB [13]) is the most closely related to our proposed MangaVQA. While MangaUB can be considered a simple VQA benchmark, it contains only eight predefined question types—such as identifying the number of characters, the weather, or the time of day—thus offering limited question diversity. As a result, MangaUB does not address a broad spectrum of VQA problems centered on text understanding in manga. Furthermore, its scope is restricted to the panel level.

In contrast, MangaVQA goes beyond individual panels and focuses on two-page spreads, reflecting how humans naturally read manga. It features diverse VQA questions grounded in textual content at the spread level, aiming to approximate the reading experience of human readers. In this regard, MangaVQA is conceptually aligned with TextVQA [27] and DocVQA [19], as it requires models to understand and reason over text embedded in images.

## 3 The Manga109 Dataset and Our Consolidated MangaOCR Dataset

This section presents the widely used manga dataset Manga109 [20] and our MangaOCR Benchmark.

### 3.1 Manga109: A Widely Used Dataset for Manga Research

Among the many comic datasets introduced in the Related Work, We selected Manga109 for its open-access license, diverse manga titles, and rich annotations and meta-information. It has also been widely used in previous comic-related research [24, 26, 3, 15, 13], making it a reliable and practical dataset for our study.

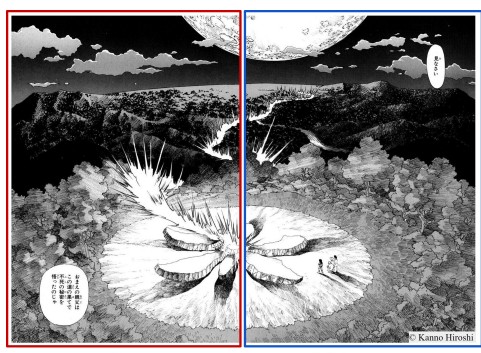

Left page      Right page

Figure 2: Illustration of a two-page spread from the Manga109 dataset.

Manga109 is a dataset composed of 109 volumes of Japanese comics (manga). Manga is a unique visual storytelling medium characterized by spatially arranged panels and artistic expression. The Manga109 dataset captures many distinctive features of manga, including its predominantly black-and-white artwork, two-page spreads, right-to-left reading order, vertical text layout, and the frequent use of stylized onomatopoeia (e.g., Boom, Bang) integrated into the illustrations. It also contains culturally specific dialogue, often incorporating honorifics and idiomatic expressions. Although these characteristics are not explicitly annotated, they present unique challenges for manga understanding tasks. Given these characteristics, Manga109 serves as a representative dataset for developing and evaluating manga understanding models. Figure 2 shows an example of two-page spreads from the Manga109 dataset.

### 3.2 MangaOCR: A Consolidated Dataset for Manga Text Recognition

Text in manga carries essential narrative information, appearing as speech balloons and stylized onomatopoeia integrated into the artwork. Recognizing such text is crucial for machine understanding

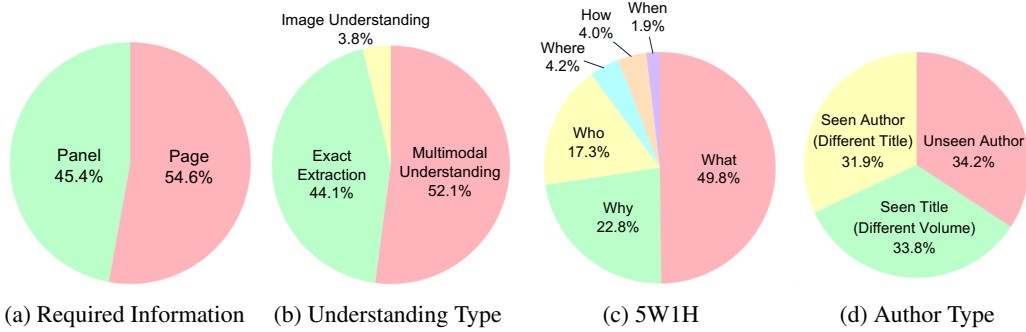

(a) Required Information    (b) Understanding Type    (c) 5W1H    (d) Author Type

Figure 3: **Distributions in MangaVQA.** The dataset is structured along four key axes: (a) Required Information, (b) Understanding Type, (c) 5W1H, and (d) Author Type.

of manga, as humans also rely on this information to comprehend the story. MangaOCR addresses this challenge by targeting two key categories of embedded text: dialogue and onomatopoeia. We construct the MangaOCR dataset by consolidating existing annotations from the Manga109 dataset and the manga onomatopoeia dataset [3]. It contains approximately 209K narrative text instances, spanning a wide variety of visual styles and layouts. Training with MangaOCR can improve the ability of LMMs to extract and interpret textual information in manga, contributing to better overall understanding. The MangaOCR task is performed on two-page spreads and primarily consists of two sub-tasks: text detection, which localizes textual regions, and text recognition, which reads the localized text.

**Author-Aware Dataset Split.** We adopt the dataset split protocol from prior work [3], with a few modifications. In the original split, the 109 volumes were divided into training, validation, and test sets based on author information. To evaluate intra-series generalization, five of the ten test volumes belong to the same series as those in the training set, where the first volume is included in the training set and the last volume is in the test set. This setting tests whether a model trained on the beginning of a series can generalize to its later volumes. To evaluate intra-author generalization, the remaining five test volumes are titles by authors who also have other works in the training set. This allows us to assess whether a model can generalize across different works by the same author.

Table 1: **Statistics of manga datasets.** More details about MangaVQA are presented in §4 and §5.

| Count type | Total | Train | Valid | Test |
|---|---|---|---|---|
| Comic volumes | 109 | 89 | 7 | 13 |
| Images | 10,602 | 8,763 | 673 | 1,166 |
| MangaOCR | | | | |
|   Dialogue | 148K | 120K | 9K | 18K |
|   Onomatopoeia | 61K | 50K | 4K | 7K |
|   Total | 209K | 170K | 13K | 26K |
| MangaVQA | | | | |
|   QA pairs | 42,421 | 41,895 | – | 526 |

To further evaluate out-of-distribution generalization with respect to author identity, we move three volumes from the validation set to the test set. These volumes are authored by individuals who did not contribute to any works in the training set. Table 1 shows the dataset statistics after the split.

## 4    MangaVQA: A Novel Benchmark for Multimodal Context Understanding

To evaluate model performance under realistic conditions, we manually created a set of question–answer (QA) pairs based on images from Manga109. Five annotators from the authors have created a high-quality evaluation set for MangaVQA. To ensure a more robust and unambiguous evaluation, we focused on questions with definite answers, avoiding those that could be inferred merely from the vague impressions of the image.

As shown in Figure 3, the question types are designed based on four key axes: (a) whether solving the question requires information from individual panels or the entire page, (b) what type of manga understanding is necessary to answer the question correctly, (c) 5W1H: whether the question asks about a person (who), an object or action (what), a time (when), a place (where), a reason (why), or a method or condition (how), and (d) inclusion of the author / title in the training split.

**(1) Exact Extraction**

Q. 風子ちゃんがもらったお人形の名前は何ですか？
(What is the name of the doll that Fuko-chan received?)

A. ふうちゃん (**Fu-chan**)

**(2) Multimodal Understanding**

Q. 捕手は、打者のどのような変化に気づきましたか？
(What changes did the catcher notice in the batter?)

A. 以前は**オープン気味に立って**いたが、今は**クローズドスタンドで立って**いる (He used to **stand with an open stance**, but now he **stands with a closed stance**.)

**(3) Image Understanding**

Q. 右下の男性は何に向かってアタックしようとしたのか？
(What was the man in the bottom right attacking?)

A. 赤ちゃん (**Baby**)

Figure 4: **Main categorization of MangaVQA questions.** MangaVQA consists of (1) Exact Extraction, where the answer is directly extracted from the image; (2) Multimodal Understanding, where the answer requires comprehension of the story beyond simple extraction; and (3) Image Understanding, which can be answered without referring to the text.

We illustrate examples along axes (b) type of manga understanding in Fig. 4. The categorization of (b) the type of manga understanding is as follows:

**(1) Exact Extraction (232 questions): Questions that Require Extracting Answer Words from the Image.** These questions necessitate accurately retrieving the answer word from the manga page. We include one example in the left of Fig. 4. The question is "風子ちゃんがもらったお人形の名前は何ですか？" ("What is the name of the doll that Fuko-chan received?") and the answer is "ふうちゃん" ("Fu-chan"), which is directly written in the dialogue. This category assesses the LMM's basic comprehension ability to identify and extract the correct answer part from the manga panels.

**(2) Multimodal Understanding (274 questions): Questions that Require the Content Comprehension in the Images.** These questions go beyond simple answer word extraction and require comprehending the context within the manga. We include one example in the middle of Fig. 4. The question is "What changes did the catcher notice in the batter?". The correct answer is "He used to stand with an open stance, but now he stands with a closed stance.". This category allows us to evaluate whether the LMM can not only recognize the dialogue but also understand its underlying meaning in the context of the narrative.

**(3) Image Understanding (20 questions): Questions Solvable without Referring to the Text in the Image.** Finally, we designed a small set of questions that can be answered without referring to the text within the images. We include one example on the right of Fig. 4. The question is "What was the man in the bottom right corner attempting to attack?". The answer is "Baby". This category relies purely on the visual depiction of characters and their actions, allowing the LMMs to infer the correct answer even in the absence of dialogue. We consider that including such questions provides a broader assessment of the LMM's capability for the manga understanding.

## 5 MangaLMM: A Specialized Model for MangaOCR and MangaVQA

We develop MangaLMM, a specialized model designed to read and understand manga in a human-like manner. To build MangaLMM, we finetune the open-source LMM Qwen2.5-VL [4] on the MangaOCR and MangaVQA datasets, resulting in a joint model for both tasks. In this section, we describe the training data construction and training details for MangaLMM.

### 5.1 Training Data Construction

**OCR Training set** $T_{OCR}$. For the OCR task, we use the MangaOCR training set, as described in §3.2. For each image, we format the sequence of text annotations as {"bbox_2d":coordinates$_1$, "text_content":text$_1$},{"bbox_2d":coordinates$_2$, "text_content":text$_2$},..., where coordinates$_i$ corresponds to the location of the text$_i$ in the image represented as $x_{top\_left}, y_{top\_left}, x_{bottom\_right}, y_{bottom\_right}$.

**Synthetic VQA training set** $T_{VQA}$**.** For the VQA task, we generate synthetic training data using GPT-4o [12](gpt-4o-2024-11-20). Following the synthetic data construction used in LLaVA [16], we generate five questions per image using both the image and its annotation from the OCR training set $T_{OCR}$. Here we exclude $< 0.1\%$ of the images where the text annotation is not included or GPT-4o refused to respond (e.g., due to violent content). As a result, we created a total of 41,895 synthetic VQA samples from 8,379 images. The prompt used for question generation is provided in the supplementary materials. We plan to release this as a training split of our MangaVQA.

## 5.2 Training Details

**LMM Selection.** Our tasks require an open-source multilingual LMM that can handle Japanese and also has strong Japanese OCR capabilities, which are important for understanding manga. Several powerful multilingual LMMs have been proposed recently [35, 31, 4, 17, 7, 21]. Among them, the Qwen series [31, 4] and Phi-4 [21] are especially notable for their Japanese OCR performance. In this work, we build MangaLMM based on Qwen2.5-VL [4], which is one of the strongest open-source models in this category.

**Training Strategy.** We perform continual finetuning on both $T_{OCR}$ and $T_{VQA}$ using the pretrained Qwen2.5-VL 7B (Qwen2.5-VL-7B-Instruct). Most hyperparameters follow the original Qwen2.5-VL configuration, with a few modifications. For Manga109 images (1654×1170 resolution), we follow Qwen2.5-VL's image resizing mechanism, which is based on pixel count thresholds, where the minimum and maximum number of input pixels are 3,136 and 2,116,800, respectively.

**Elapsed Time for Training.** Each dataset is trained for one epoch. Training Qwen2.5-VL 7B using four NVIDIA A100 GPUs took about 1 hour when using $T_{OCR}$ or $T_{VQA}$, and about 2 hours when using both $T_{OCR}$ and $T_{VQA}$.

# 6 Experiments

**Evaluation Protocol for MangaOCR.** We follow the evaluation protocols from prior OCR studies [33, 11] and ICDAR 2019 multilingual OCR competitions [6, 36, 29, 22]. First, a predicted bounding box is considered a correct detection if its intersection over union (IoU) with a ground truth box exceeds 0.5. Based on the matched boxes, we compute precision (P), recall (R), and the harmonic mean (Hmean). Second, for each matched box, we calculate the normalized edit distance (NED) between the predicted and ground truth texts as a character-level metric. NED ranges from 0 to 1, with higher values indicating better performance; details are in the supplementary materials.

Since LMMs sometimes output the same word repeatedly, we apply post-processing to exclude repeated text segments that appear more than ten times, treating them as noise. Except for the analysis in § 6.3, we report only the end-to-end Hmean for simplicity.

**Evaluation Protocol for MangaVQA.** Following LLaVA-Bench [16], we adopt the LLM-as-a-judge approach [37] as our evaluation metric. We provide GPT-4o [12] (gpt-4o-2024-11-20) with the question, a human-written answer, and the model's response. Based on the human-written answer, GPT-4o assesses whether the model's response is appropriate and relevant to the question, using a 1–10 scale. The prompt used for LLM-as-a-judge is provided in the supplementary materials.

**LMMs Used for Comparison.** We evaluate two proprietary LMMs, gpt-4o-2024-11-20 [12] and gemini-2.5-flash-preview-04-17 [9], and two open-source LMMs, Phi-4-multimodal-instruct [1] and Qwen2.5-VL-7B-Instruct [4].

## 6.1 Main Results

Table 2 compares LMMs for both MangaOCR and MangaVQA tasks. Overall, MangaLMM can handle both tasks effectively: it achieves over 70% OCR score and outperforms GPT-4o in VQA score (5.75 vs. 6.57).

**Analysis of Low Performance on MangaOCR.** As shown in Table 2, GPT-4o, Gemini 2.5, Phi-4, and Qwen2.5-VL all show near-zero score on the MangaOCR benchmark. Most of their predictions consist of meaningless repetitions or short repeated tokens. The extremely low OCR score before finetuning is likely due to two main factors: (1) these models are not familiar with manga data, and

Table 2: **Comparison of LMMs on MangaOCR and MangaVQA.**

| Method | MangaOCR Hmean (%) | MangaVQA LLM (/10.0) |
|---|---|---|
| GPT-4o | 0.0 | 5.76 |
| Gemini2.5 Flash | 0.0 | 3.87 |
| Phi-4-Multimodal | 0.0 | 3.08 |
| Qwen2.5-VL 7B | 0.9 | 5.36 |
| MangaLMM (Ours) | **71.5** | **6.57** |

Table 3: **Effect of finetuning (FT).** FT is performed on the OCR training set $T_{OCR}$, the VQA training set $T_{VQA}$, or both.

| FT data | MangaOCR Hmean (%) | MangaVQA LLM (/10.0) |
|---|---|---|
| None | 0.9 | 5.36 |
| $T_{OCR}$ | **74.9** | 1.03 |
| $T_{VQA}$ | 0.0 | 6.46 |
| $T_{OCR}+T_{VQA}$ | 71.5 | **6.57** |

(2) their weak detection capabilities may limit OCR performance. Prior work [32] has shown that GPT-4o, for example, exhibits poor detection ability, which may also apply to the other models.

Despite the near-zero OCR score—where not only position information is missing but even the correct text content is not generated—these models still manage to answer certain VQA questions that require interpreting text within the image. This is somewhat *counterintuitive*. Although the models fail to explicitly output the correct OCR results, they appear to capture some textual semantics from the image. This suggests that they are able to extract relevant information needed for answering VQA questions, even without performing OCR correctly.

**Analysis of the Effect of Finetuning.** Table 3 shows the effect of finetuning. Finetuning Qwen2.5-VL on $T_{OCR}$ and $T_{VQA}$ allows the model to specialize in each respective task. On MangaOCR, the finetuned model achieves a significant improvement to a score of 74.9%, which we provide more interpretation in § 6.3. On MangaVQA, while the model initially underperforms compared to GPT-4o, it demonstrates a notable performance gain, even surpassesing GPT-4o. These results highlight the effectiveness of our synthetic VQA training set $T_{VQA}$, which we further analyze in §6.4.

**Analysis from the Perspective of Task Interference.** MangaLMM, a Qwen2.5-VL model fine-tuned jointly on both $T_{OCR}$ and $T_{VQA}$, shows a slight drop in OCR performance compared to using $T_{OCR}$ alone, but achieves a small gain in VQA score over using $T_{VQA}$ alone. A common issue in multi-task learning is *task interference* [18, 34, 8, 5], where models jointly trained on multiple tasks (e.g., $A$ and $B$) tend to perform worse on task $A$ compared to models trained solely on $A$. Under this assumption, one might expect the VQA performance of a jointly trained OCR+VQA model to degrade relative to a VQA-only model. Interestingly, we observe a slight improvement in VQA score under joint training, contrary to typical interference expectations. This suggests that although task interference may be present, the enhanced OCR capability likely provides beneficial textual cues that marginally improve VQA performance.

## 6.2 Effect of Model and Dataset Size

Table 4 shows the performance of Qwen2.5-VL models of different sizes (3B and 7B) under various finetuning settings. Similar to the 7B model, the 3B model shows a slight drop in MangaOCR performance when finetuned on both $T_{OCR}$ and $T_{VQA}$, while its MangaVQA performance improves slightly. Table 5 shows the results of varying dataset size (25%, 50%, 75%, and 100%). We observe that performance generally improves as the dataset size increases.

## 6.3 Performance Analysis of MangaOCR

Table 6 shows MangaOCR performance at both the detection and end-to-end stages. The Hmean of detection is 75.8%, while the Hmean of end-to-end reaches 68.7%, implying that once text regions are detected, the model can read them with approximately 90% (=68.7 / 75.8) accuracy. Some false positives occur when the model predicts text that is indeed present in the manga but not included in the annotations—for example, page numbers or editorial marks that are not part of the narrative content such as dialogue or onomatopoeia. As a result, the precision is unlikely to reach 100%. Compared to precision, recall is relatively low (65.0%). This suggests that around 35% of ground-truth narrative text remains undetected, indicating room for improvement in capturing all semantically relevant content. Qualitative analysis of MangaOCR is provided in the supplementary materials.

Table 4: Effect of model size (3B and 7B).

| Size | FT data | MangaOCR Hmean (%) | MangaVQA LLM (/10.0) |
|---|---|---|---|
| 3B | None | 0.1 | 4.30 |
| | $T_{OCR}$ | 73.5 | 3.78 |
| | $T_{VQA}$ | 0.0 | 5.71 |
| | $T_{OCR}+T_{VQA}$ | 66.5 | 5.86 |
| 7B | None | 0.9 | 5.36 |
| | $T_{OCR}$ | **74.9** | 1.03 |
| | $T_{VQA}$ | 0.0 | 6.46 |
| | $T_{OCR}+T_{VQA}$ | 71.5 | **6.57** |

Table 5: Effect of dataset size.

| Ratio (%) | MangaOCR Hmean (%) | MangaVQA LLM (/10.0) |
|---|---|---|
| 25 | 59.0 | 6.15 |
| 50 | 64.9 | 5.99 |
| 75 | 68.4 | 6.39 |
| 100 | 71.5 | 6.57 |

Table 6: Detection and end-to-end performance on MangaOCR.

| Stage | Prec. | Recall | Hmean |
|---|---|---|---|
| Detection | 80.3 | 71.8 | 75.8 |
| End-to-end | 72.8 | 65.0 | 68.7 |

## 6.4 Performance Analysis of MangaVQA

**Category-wise VQA Performance.** Figure 5 shows a breakdown of model performance across the annotated categories in MangaVQA. We observe performance improvements across nearly all tags in every annotated category, indicating that our training contributes to a consistent and balanced enhancement in VQA capabilities. For example, perhaps surprisingly, the model generalizes well to questions from unseen authors, although the performance gain is slightly smaller compared to other tags (rightmost figure).

The only exception is the questions that do not require textual information ("Understanding Type = Image"). In this case, a slight performance drop has been observed after training. We hypothesize this is because our training is strongly text-aware — not only is the model trained on MangaOCR, but synthetic VQA generation is guided with text annotation. We do not consider this a major limitation as uniqueness of manga lies in its multimodality and use cases on non-textual understanding are relatively rare. Still, the training methods better suited for such cases is left for future work.

**Effect of OCR Annotation when Generating VQA Data.** On creating synthetic QA pairs for training, we provide GPT-4o with the OCR annotation as part of the prompt. Here, we ablate the impact of this by comparing the effect of VQAs made with and without text annotation. As shown in Table 7, the performance of a model on VQA data generated without OCR information (5.44) does not outperform GPT-4o's own score (5.76). In contrast, OCR-guided VQAs substantially improve the score (6.57), even outperforming the GPT-4o. These results suggest that OCR annotations help GPT-4o generate high-quality QA pairs beyond its inherent performance.

Table 7: Effect of OCR Annotation on VQA Generation.

| OCR Annot. | LLM (/10.0) |
|---|---|
| | 5.44 |
| ✓ | **6.57** |

**Qualitative Analysis for MangaVQA.** In Figure 6, we provide a few examples comparing the outputs of the original Qwen model and our trained model. Here, we briefly summarize our observations: **Left**: The original model generates a general answer based on the panel in which the person in question appears, while the trained model's answer is based on the content of a text bubble and is more specific, resulting in a score increase of 7 ($3 \rightarrow 10$). **Middle**: The original model extracts text irrelevant to the question, while the trained model extracts the correct text, resulting in a score increase of 8 ($2 \rightarrow 10$). **Right**: The original model extracts the wrong dish name, which is not asked about in the question. The trained model correctly identifies the target dish name but fails to extract it character by character, resulting in no score improvement ($2 \rightarrow 2$).

## 7 Conclusion and Discussion

We present MangaVQA, a benchmark for evaluating to what extent LMMs can understand manga in a human-like way through contextual visual question answering, and MangaOCR, a consolidated benchmark for in-page text recognition. Together, they cover both textual and narrative aspects of multimodal manga understanding. To establish a strong baseline, we develop MangaLMM, a specialized model jointly finetuned on OCR and VQA tasks. Experiments show that even state-of-the-

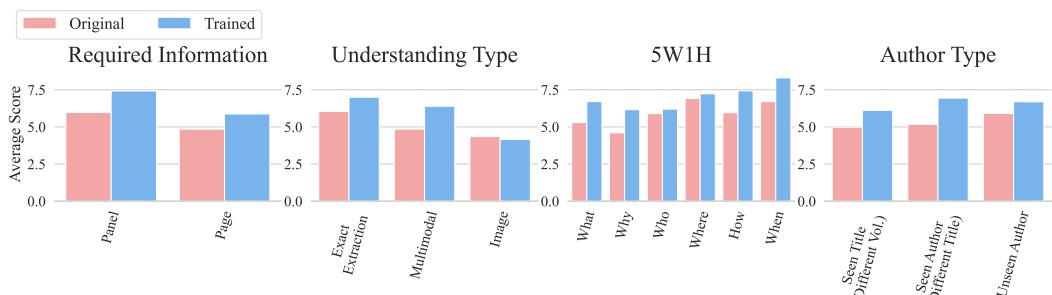

Figure 5: **Category-wise score breakdown.** Compared to the original model (Qwen2.5-VL-7B-Instruct), our trained MangaLMM improves scores across nearly every tag in every category.

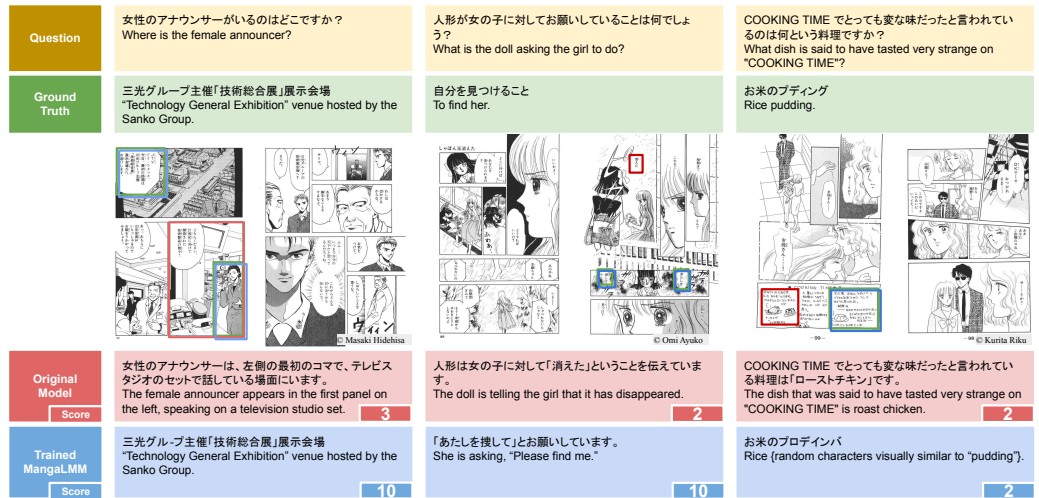

Figure 6: **Qualitative analysis on MangaVQA.** The regions in the image relevant to the question or models' answer are highlighted with boxes in corresponding colors. In the left and middle examples, the model's performance improves significantly after training, whereas in the right example, the trained model still struggles to produce an accurate answer.

art proprietary LMMs struggle with manga's unique complexity, while MangaLMM performs well across both tasks. By releasing open benchmarks, synthetic data, and a strong open-source baseline, we aim to advance research in multimodal manga understanding.

**Limitation.** One limitation of our model is its slow inference speed for OCR. LMMs are much slower than dedicated OCR models; for instance, processing 1,166 test images with 25,651 texts takes several hours on an A100 GPU. In contrast, a dedicated OCR model like DeepSolo [33], running at over 10 FPS, would finish in about 2 minutes. This slowdown stems from the large number of output tokens and occasional repeated or looping outputs during inference.

**Impact Statement.** Copyright issues surrounding manga data are often complex. In the case of PoPManga [25], its training data is not publicly available, and its test data is inaccessible from several Asian countries due to copyright restrictions. In contrast, the Manga109 [20] dataset we use consists only of works for which explicit permission for research use has been obtained from the manga authors. We hope that future research in the manga domain will increasingly rely on copyright-clear datasets like Manga109, enabling the field to advance in a cleaner and more reliable manner.

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
