# OpenReview forum: "MangaVQA and MangaLMM: A Benchmark and Specialized Model for Multimodal Manga Understanding"
_NeurIPS.cc/2025/Datasets_and_Benchmarks_Track — Submitted to NeurIPS 2025 Datasets and Benchmarks Track_

### Official Review · Reviewer_hbJq · 2025-06-30

**Rating:** 4
**Confidence:** 4

**Summary:**

This paper introduces two benchmarks MangaOCR ****and MangaVQA ****to evaluate multimodal understanding of manga, with a focus on in-image text recognition and VQA. The authors further propos MangaLMM, a Qwen2.5-VL-based model finetuned on these benchmarks. The paper demonstrates that proprietary LMMs (like GPT-4o and Gemini 2.5) struggle on these tasks, whereas MangaLMM performs well due to specialized training.

**Dataset Code Accessibility:**

Yes

**Dataset Code Comments:**

These benchmarks are thoughtfully designed and fill a clear gap in the community.

**Ethical Considerations:**

No, there are no or only very minor ethics concerns

**Final Justification:**

The authors' rebuttal adequately addressed my primary concerns. Specifically:

(1) Dataset Size: The authors provided convincing comparisons to other high-quality, domain-specific benchmarks of similar or smaller scale, clarifying that MangaVQA’s size is reasonable for its purpose and well-curated.

(2) Empirical Reliability: The additional runs with mean and standard deviation confirmed the stability of reported results, which enhances confidence in the findings.

Given that these were my main concerns, and they have been resolved, I have increased my score. I have no remaining major issues at this time.

**Limitations Weaknesses:**

1. While the 526 QA pairs in MangaVQA are high quality, the scale is relatively small compared to modern VQA datasets.
2. Even partial replication of key experiments (e.g., three seeds for MangaLMM) would greatly improve the paper’s empirical reliability.

**Strengths Contributions:**

The authors contribute two high-quality benchmarks tailored for manga: *(1) MangaOCR* targets real-world Japanese text recognition embedded in images (e.g., vertical layout, onomatopoeia, etc.). *(2) MangaVQA* features 526 manually curated QA pairs grounded in two-page manga spreads, covering 5W1H and multimodal comprehension.

---

> ### Author Rebuttal · Authors · 2025-07-30
>
> We are grateful for the reviewer’s positive comments and recognition of the following aspects of our work:
>
> **S1.** Contribute two high-quality benchmarks tailored for manga: (1) MangaOCR for real-world Japanese manga text, and (2) MangaVQA for multimodal manga understanding.
>
> Below, we present detailed responses to the reviewer’s comments.
>
> ### W1: Dataset size is relatively smaller than other modern VQA datasets
>
> ---
>
> We acknowledge that there are larger general-purpose VQA benchmarks—such as MMMU [a] (11.5k total, 900 for test) and MMBench [b] (3k total, 1.8k for test). However, domain-specific benchmarks often prioritize quality over scale. While smaller in size, they are widely regarded as valuable tools for evaluating specialized capabilities. Several well-regarded examples include:
>
> - VisIT-Bench [c] (592 samples) – Real-world instruction-following tasks [NeurIPS D&B 2023]
> - MM-Vet [d] (218 samples) – Complex visual reasoning [ICML 2024]
> - V*Bench [e] (191 samples) – High-resolution understanding [CVPR 2024]
> - JMMMU [f] handmade subset (600 samples) – Japanese cultural context [NAACL 2025]
>
> These benchmarks, despite their modest scale, are widely used because they offer sufficient discriminative power in specialized domains. Likewise, MangaVQA consists of 526 carefully curated examples. Each question was curated by the seven authors to ensure clarity and avoid subjective interpretation or ambiguous answers. This rigorous process has resulted in a high-quality benchmark that offers consistent and reliable evaluation for the manga domain.
>
> We believe that, as demonstrated in these prior works, 500+ well-curated samples can provide a sufficient evaluation signal for targeted domains. Building on this, MangaVQA’s focused scope and careful construction make it a valuable benchmark for evaluating model performance in the unique domain of manga.
>
> [a] Yue et. al, MMMU: A Massive Multi-discipline Multimodal Understanding and Reasoning Benchmark for Expert AGI, CVPR2024
>
> [b] Liu et. al, MMBench: Is Your Multi-modal Model an All-around Player?, ECCV2024
>
> [c] Bitton et. al, VisIT-Bench: A Benchmark for Vision-Language Instruction Following Inspired by Real-World Use, NeurIPS D&B 2023
>
> [d] Yu et. al, Mm-vet: Evaluating large multimodal models for integrated capabilities, ICML2024
>
> [e] Wu et. al, V*: Guided Visual Search as a Core Mechanism in Multimodal LLMs, CVPR2024
>
> [f] Onohara et. al, JMMMU: A Japanese Massive Multi-discipline Multimodal Understanding Benchmark for Culture-aware Evaluation, NAACL2025
>
> &nbsp;
>
> ### W2: Replication of key experiments to improve empirical reliability
>
> ---
>
> In response to the reviewer’s helpful suggestion, we replicated the main experiments reported in Table 3, running each configuration three times. The table below presents the mean and standard deviation across these runs. The overall performance trends remain consistent, and this additional evaluation improves the empirical reliability of our results. We appreciate the reviewer’s constructive feedback and will incorporate these results into the revised version of the paper.
>
> | FT data | MangaOCR  | MangaVQA|
> |---|---|---|
> | T_OCR | 74.5 ± 1.3 | 1.39 ± 0.64 |
> | T_VQA | 0 ± 0 | 6.53 ± 0.12 |
> | T_OCR + T_VQA (MangaLMM) | 71.2 ± 0.6 |  6.70 ± 0.03  |
>
> Note: The MangaVQA score reported here differs slightly from that in the main paper. As pointed out by Reviewers y1Zw and cc3P, using GPT-4o as the evaluation judge may introduce circular bias. Therefore, in this rebuttal, we report results based on evaluations using Gemini 2.5 Flash as an independent judge.

---

> > ### Comment · Reviewer_hbJq · 2025-08-05
> > **Official Comment by Reviewer hbJq**
> >
> > Thank you for the response. It addressed my concerns. I have updated my score accordingly.

---

> > > ### Author Response · Authors · 2025-08-05
> > >
> > > Thank you for your response and for updating your score. We're pleased that our rebuttal addressed your concerns, and we appreciate your thoughtful evaluation.

---

### Official Review · Reviewer_w91Z · 2025-07-01

**Ethics Flags:** Data privacy, copyright, and consent
**Rating:** 5
**Confidence:** 3

**Summary:**

This paper presents two benchmarks for multimodal manga understanding including OCR and VQA. In addition, a manga-specialized LMM is developed for this specific task. Extensive experimental results and ablatio studies demonstrate the necessity and effectiveness of the proposed benchmarks and specialized LMM.

**Additional Feedback:**

See weaknesses.

**Dataset Code Accessibility:**

Yes

**Dataset Code Comments:**

NA

**Ethical Considerations:**

No, there are no or only very minor ethics concerns

**Final Justification:**

Thanks for providing the additional experimental results, which address most of my concerns. I keep my initial rating.

**Limitations Weaknesses:**

A major limitation is that the LMM is fine-tuned and evaluated specifically for manga-related tasks, rather than being co-trained to retain common knowledge of natural images. I wonder if it is possible to develop an LMM capable of handling both manga and natural image understanding, which would be more convincing. It's required to show evaluation results on natural image benchmarks to compare with general-purpose baselines like GPT-4o and the Qwen-VL series. Alternatively, you could fine-tune the model on a combined dataset that includes natural image understanding data.

**Strengths Contributions:**

1. This paper is well-organized and easy to follow.
2. The proposed evaluation benchmarks are necessary to comprehensively evaluate existing LMMs for manga multimodal understanding.
3. The experimental results and ablation stuides are convincing.

---

> ### Author Rebuttal · Authors · 2025-07-30
>
> We greatly appreciate the reviewer’s positive feedback and are pleased that the following contributions were acknowledged:
>
> **S1.** A well-organized and easy-to-follow presentation.
>
> **S2.** Necessary benchmarks for evaluating LMMs on multimodal manga understanding.
>
> **S3.** Convincing results and ablation studies.
>
> Below, we respond to the reviewer’s comment.
>
> ### Exploring the Capability of LMMs for Manga and Natural Image Understanding
>
> ---
>
> We appreciate the reviewer’s insightful suggestion to explore models that can handle both manga and natural images. To explore this direction, we conducted additional evaluations as suggested.
>
> **Suggestion1. Show evaluation results on natural image benchmarks.**
>
> To address this point, we evaluated general-purpose baselines (GPT-4o, Gemini 2.5 Flash, Phi-4-Multimodal, and Qwen2.5-VL-7B) as well as our MangaLMM on two representative benchmarks commonly used for assessing LMMs’ performance on natural images: MMMU [a] and MMBench [b].
> The table below (an extended version of Table 2) shows the results.
> Scores for models other than our MangaLMM are obtained from the official MMMU leaderboard or the Phi-4 technical report [c].
> As expected, the general-domain performance of MangaLMM (fine-tuned from Qwen2.5-VL-7B) drops noticeably due to domain specialization, underscoring the challenge of maintaining broad visual understanding after task-specific fine-tuning.
>
> | Method | MangaOCR | MangaVQA  |  MMMU | MMBench |
> |---|---|---|---|---|
> | GPT-4o | 0.0 | 5.88 | 70.7 | 89.0 |
> | Gemini 2.5 Flash | 0.0 | 3.85 | 79.7 | - |
> | Phi-4-Multimodal | 0.0 | 3.61 | 55.1 | 86.7 |
> | Qwen2.5-VL 7B | 0.9 | 5.51 | 58.6 | 87.8 |
> | MangaLMM | 71.5 | 6.68 | 25.8 | 1.5 |
>
> [a] Yue et. al, MMMU: A Massive Multi-discipline Multimodal Understanding and Reasoning Benchmark for Expert AGI, CVPR2024
>
> [b] Liu et. al, MMBench: Is Your Multi-modal Model an All-around Player?, ECCV2024
>
> [c] Abdin et. al, Phi-4 technical report, arXiv2024
>
> &nbsp;
>
> **Suggestion2. Fine-tune the model on a combined dataset that includes natural image understanding data.**
>
> To explore this suggestion, we fine-tuned our model on a combined dataset consisting of manga data and 50K randomly sampled natural image examples from the LLaVA-OneVision dataset [d] . As shown in the table below (an extended version of Table 3), this joint fine-tuning substantially restores the model’s performance on natural image benchmarks, while preserving strong performance on our MangaOCR and MangaVQA benchmarks.
>
> | FT data | MangaOCR | MangaVQA  |  MMMU | MMBench |
> |---|---|---|---|---|
> | T_OCR + T_VQA (MangaLMM)  | 71.5 | 6.68 | 25.8 | 1.5  |
> | T_OCR + T_VQA + LLaVA onevision 50K  | 70.2 |  6.44 |  49.6 | 82.0 |
>
> [d] Li et al., LLaVA-OneVision: Easy Visual Task Transfer, arXiv2024
>
> These results demonstrate that, although domain specialization may reduce general capability, it can be effectively recovered through joint training—highlighting the feasibility of developing LMMs that are both manga-capable and broadly applicable to natural image understanding.
>
> Note: The MangaVQA score reported here differs slightly from that in the main paper. As pointed out by Reviewers y1Zw and cc3P, using GPT-4o as the evaluation judge may introduce circular bias. Therefore, in this rebuttal, we report results based on evaluations using Gemini 2.5 Flash as an independent judge.

---

> > ### Comment · Reviewer_w91Z · 2025-08-05
> >
> > Thanks for providing the additional experimental results, which address most of my concerns. I keep my initial rating.

---

> > > ### Author Response · Authors · 2025-08-05
> > >
> > > Thank you for your response. We're glad that the additional results addressed most of your concerns, and we appreciate your positive rating.

---

### Official Review · Reviewer_cc3P · 2025-07-03

**Rating:** 5
**Confidence:** 4

**Summary:**

The paper proposes two copyright‑cleared benchmarks for machine understanding of Japanese comics: MangaOCR (≈209 k dialogue / onomatopoeia boxes) and MangaVQA (526 manually written question–answer pairs) built on the public Manga109 corpus. The authors then finetune Qwen‑2.5 VL to create MangaLMM, a 7 B‑parameter vision–language model that jointly tackles OCR and VQA, trained with real OCR labels and 41 k synthetic VQA examples. On MangaOCR, MangaLMM lifts end‑to‑end Hmean from <1 % (base model) to 71.5 %, and on MangaVQA it beats GPT‑4o under an LLM‑as‑judge protocol (6.57 vs 5.76) while proprietary models fail almost completely at OCR . All code, data splits and training scripts are promised for open release, positioning the work as a strong baseline for multimodal manga research

**Dataset Code Accessibility:**

Yes

**Ethical Considerations:**

No, there are no or only very minor ethics concerns

**Final Justification:**

I have read the author’s response. They address my concerns. So I decide to maintain the acceptance decision.

**Limitations Weaknesses:**

(1) Evaluation leans on GPT‑4o as an automatic judge without any human VQA scoring, which risks circular bias.
(2) The synthetic VQA training data are generated by the same GPT‑4o family, so gains may partly reflect prompt‑domain alignment rather than real understanding.
(3) Inference is slow (hours for 1 k pages) and recall still misses ~35 % of narrative text, limiting practical deployment.

**Strengths Contributions:**

(1) Delivers the first paired OCR + VQA benchmark for manga, with careful author‑aware train/val/test splits to probe generalisation.
(2) Supplies an open, reasonably sized 7 B model that materially outperforms larger commercial LMMs on domain tasks, highlighting the value of domain‑specific finetuning.
(3) Provides detailed ablations on model size, dataset size and multi‑task interference, giving future work useful reference points.

---

> ### Author Rebuttal · Authors · 2025-07-30
>
> We sincerely thank the reviewer for the encouraging feedback. We appreciate the recognition of the following contributions:
>
> **S1.** The first paired OCR+VQA benchmark for manga, with author-aware splits to assess generalization.
>
> **S2.** A finetuned open 7B model that outperforms larger commercial LMMs, highlighting the value of domain-specific finetuning.
>
> **S3.** Detailed ablations on model size, dataset scale, and task interference, providing useful reference points for future work.
>
> Below, we provide detailed responses to each of the reviewer’s comments.
>
> ### W1. Circular Bias Risk and Human Scoring
>
> ---
>
> We acknowledge the potential risk of circular bias in the MangaVQA benchmark.
> In particular, evaluating GPT-4o’s responses using GPT-4o itself may introduce such bias.
> To eliminate this risk, we used Gemini 2.5 Flash as an independent judge for evaluation.
> The results are shown in the tables below. These tables are extended versions of Tables 2 and 3 in the main paper. Even with this alternative judge, the overall trend remains consistent with the original results using GPT-4o as a judge, indicating that our conclusions still hold.
>
> **Comparison of LMMs (extended version of Table 2)**
>
> | Method | Judge by GPT-4o  | Judge by Gemini 2.5 Flash |
> |---|---|---|
> | GPT-4o | 5.76 | 5.88 |
> | Gemini 2.5 Flash | 3.87 | 3.85 |
> | Phi-4-Multimodal | 3.08 | 3.61 |
> | Qwen2.5-VL 7B | 5.36 | 5.51 |
> | MangaLMM (ours) | 6.57 | 6.68 |
>
> **Finetuning on different training sets (extended version of Table 3):**
>
> | FT data | Judge by GPT-4o  | Judge by Gemini 2.5 Flash |
> |---|---|---|
> | T_OCR | 1.03 | 1.03 |
> | T_VQA | 6.46 | 6.43 |
> | T_OCR + T_VQA (MangaLMM) | 6.57 | 6.68 |
>
> Regarding the absence of human VQA scoring, we agree that human evaluation is ideal; however, manually scoring all responses was especially costly. Instead, as shown in Section D.1 (Line 67), we sampled 100 examples from MangaVQA and conducted human evaluation. We then computed the correlation between GPT-4o’s scores and human judgments, and found a strong correlation (0.22 difference in average score, and Pearson correlation of 0.94) —suggesting that GPT-4o’s scores are reasonably aligned with human evaluations.
>
> &nbsp;
>
> ### W2. Synthetic Data and Prompt-Domain Alignment Bias
>
> ---
>
> We understand that this concern points to a related but distinct form of circular bias from W1.
> In this case, the issue lies not in GPT-4o evaluating its own responses, but in using GPT-4o both to generate synthetic VQA training data and to evaluate the model's performance—raising a risk of prompt-domain alignment rather than genuine understanding.
> We believe this concern is also addressed by evaluating MangaVQA using Gemini 2.5 Flash instead of GPT-4o, as shown in the tables above.
> Even when using a different model family as the evaluator, the performance trends remain consistent, suggesting that the observed gains are not simply due to prompt-domain alignment bias.
> To further address such risks, we plan to update all MangaVQA evaluation results in the paper using Gemini 2.5 Flash as the evaluation judge.
>
> &nbsp;
>
> ### W3. Slow Inference and Incomplete Recall in MangaOCR Limit Practical Deployment
>
> ---
>
> We acknowledge that the current inference speed and recall rate pose challenges for practical deployment, as discussed in the limitation section and Section 6.3 of our paper. While the slow inference is largely due to the limitations of current LMMs, we agree that improving both efficiency and recall remains an important direction for future work.
> We hope that our MangaLMM can serve as a strong baseline, and that MangaOCR and MangaVQA offer useful guidance for future research aiming to make LMMs more practical and reliable in real-world applications.

---

### Official Review · Reviewer_y1Zw · 2025-07-03

**Rating:** 3
**Confidence:** 5

**Summary:**

This paper introduces MangaVQA (526 human-authored question–answer pairs on two-page spreads from Manga109), MangaOCR (aligned text–box annotations) and MangaLMM, a Qwen-2.5-VL-7B model fine-tuned on MangaOCR plus ≈ 40 k GPT-4o-generated QAs.The authors claim their model outperforms GPT-4o and Gemini-2.5-Flash on their benchmark (6.57 vs 5.76 / 10) and provides the first “comprehensive” evaluation suite for manga understanding.

**Dataset Code Accessibility:**

Yes

**Ethical Considerations:**

No, there are no or only very minor ethics concerns

**Final Justification:**

I appreciate the authors clarifications and additional results. The paper is on a timely topic, with planned code/data release. However, for a dataset/benchmark paper the evidence is insufficient.

The synthetic VQA is generated from image+OCR, biasing toward text extraction; the test set is overwhelmingly text-based (≈20 image-only/526), limiting claims about multimodal understanding. Results rely on LLM-as-judge (even with Gemini as an alternate), with minimal human validation. Crucial ablations are missing (generator/prompt/judge sweeps; spread vs pages/panels/reading order; OCR-only and text-masked baselines). My quick checks show OCR-only often suffices, and panel-ordered inputs outperform spreads—suggesting formatting/OCR reliance rather than hard visual reasoning.

**I therefore maintain my borderline-reject score.** If it proceeds, add the ablations and baselines above plus script-reproducible dataset counts.

**Limitations Weaknesses:**

However, there are quite a few weankesses and limitations that are worth pointing.

The core weakness is that the work does not yet offer the ablation studies ordinarily expected for NeurIPS-level claims. All 39 k synthetic questions are authored by a single model—GPT-4o—whose performance is already not state-of-the-art among proprietary LMMs, yet the authors never check whether better generators (Gemini 2.5-Pro, o3, Claude-3, etc.) produce higher-quality or more diverse QA pairs. Likewise, the only supplementary signal fed to the generator is the raw Japanese OCR dump; no spatial tags, reading-order hints, storyline summaries, or panel metadata are supplied. Prior studies have shown that even advanced LMMs fail to follow the correct panel sequence on a single manga page—so expecting coherent reasoning over a two-page spread with nothing but unstructured text is optimistic at best. Yet the paper provides no experiments to disentangle whether the bottleneck is Japanese language proficiency, manga-specific visual complexity, or the two-page format itself. No exploration is provided in the current draft of (i) prompt variants, (ii) alternative input formats (spread vs single page vs sequential panels), or (iii) generator choice.

The following are all other weaknesses and limitations:
- tiny and unbalanced test set: 526 QAs total, only 20 require vision without text, yielding wide confidence intervals and text-dominant evaluation;
- circular evaluation: GPT-4o both creates training questions and judges test answers; headline 0.8-point gain lacks statistical significance or an independent referee;
- language limitation: all text is Japanese; models with strong English but weak JP ability are unfairly penalised, yet no bilingual version is provided;
- bounding-box noise: supplementary Table D shows that adding bbox coordinates hurts GPT-4o’s QA quality, but those tokens are still used during fine-tuning, bloating context length without benefit;
- two-page assumption untested: No empirical evidence that opposite-page context helps; digital readers commonly default to single pages.
- one-seed reporting: Every result is from a single run; no bootstrap CIs, no multiple seeds, “cost reasons” cited for skipping them.
- OCR metric rigidity: Models must emit exact JSON boxes; correct text with imprecise coordinates scores 0 %, disadvantaging models focused on semantics.
- limited stylistic coverage despite copyright free material of different style exists: The authors choose Japanese Manga-only material with the justification of Copyright free, but this limits to only B/W 1990s print manga (Manga109); no colour, vertical-scroll Webtoons, or non-Japanese works, limiting external validity. Other data sources exists (CoMix, COMICS, eBDtheque, etc.) which are out of copyright or accessible for research under request.

**Strengths Contributions:**

- The paper motivations are sound and it is easy to go through.
- The task is also timely and under-explored: Japanese manga mix complex layout, stylised glyphs and dense narrative; and pointed by recent studies [a], no proper VQA dataset exists.
- Motivation or reading (OCR) and understanding (VQA) are well fused in this study: coupling text detection with downstream reasoning is principled and mirrors real-world pipelines.
- the synthetic-data recipe is clear and reproducible: a single GPT-4o prompt yields five QAs per spread; the supplementary discloses it well.
- the paper also explored Initial human validation on 100 QA pairs and 100 judge decisions, receiving 4-annotator checks (≈ 80 % correctness, Pearson 0.94 with GPT-judge).

---

> ### Author Rebuttal · Authors · 2025-07-30
>
> We thank the reviewer for the extensive review, positive feedback, and many insightful comments. Before addressing each comment, we would first like to clarify a few potential misunderstandings.
>
> &nbsp;
>
> ### Potential Misunderstandings and Clarification
>
> ---
>
> **Potential Misunderstanding 1 (M1): Scope of the Study**
>
> One possible misunderstanding is that “this study aims to address comics in all languages.” However, as clearly stated in the first sentence of the abstract and throughout the paper, our focus is specifically on Japanese comics (manga). While multilingual extension is an interesting future direction, it is beyond the scope of this study. M1 may have influenced W1(ii), W4, W6, and W9.
>
> **Potential Misunderstanding 2 (M2): “The authors choose Japanese Manga-only material with the justification of Copyright free”**
>
> Our choice of the Manga109 dataset was not solely based on copyright. As stated in Section 3.1 (Line 95), we selected Manga109 for its diverse titles, rich annotations, and wide use in prior comic-related research. Manga109 has been cited in over 1,500 publications, reflecting its recognition as a practical resource. Its open-access license, granted through direct author permissions, also ensures legal reliability. M2 may have influenced W9.
>
> **Potential Misunderstanding 3 (M3): Main Contribution of the Study**
>
> The reviewer claims that limited ablation on synthetic VQA generation is a “core weakness.” However, our main contribution lies in constructing the MangaVQA and MangaOCR benchmarks and developing MangaLMM as a strong baseline for these tasks. Thus, focusing primarily on limitations of synthetic VQA generation deviates from our intended scope. Moreover, the strong performance of MangaLMM—surpassing GPT-4o on MangaVQA—demonstrates that our synthetic data is sufficiently effective and supports its utility as a practical baseline. M3 may have influenced W1.
>
> &nbsp;
>
> ### Detailed Response to Comments
>
> ---
>
> Based on the clarifications above, we address each of the comments below.
> Additional experiments discussed in this rebuttal will be included in the revised version.
>
> **W1: Limited ablations for synthetic VQA generation**
>
> As clarified in M3, the main focus of this work is not to optimize synthetic data, but to examine whether a model finetuned on synthetic VQA data can achieve strong performance on MangaVQA.
> Notably, MangaLMM outperforms GPT-4o, a widely used strong proprietary model, supporting the viability of our setup as a practical starting point.
>
> While these factors are beyond the main scope of our study, we address each of the reviewer’s suggestions and include an additional experiment on generator choice below.
>
> (i) Prompt variants: We tested OCR dump and spatial tags as inputs (Tables 7 and D). The OCR dump significantly improved performance, while spatial tags had little effect.
>
> (ii): Input format: Please refer to our response in W6 for clarification on the choice of two-page spread format.
>
> (iii) Generator choice: As suggested, we conducted experiments using o3 and observed similar results (see table below). To address W3, we used Gemini 2.5 Flash as the judge.
>
> | synthetic data generation model | MangaOCR  | MangaVQA |
> |---|---|---|
> | gpt-4o-2024-11-20 | 71.5 | 6.68 |
> | o3-2025-04-16 | 70.7 | 6.73 |
>
> &nbsp;
>
> **W2-1: The size of the MangaVQA (526 QAs) is tiny.**
>
> We acknowledge that there are larger general-purpose VQA benchmarks—such as MMMU (11.5k total, 900 for test) and MMBench (3k total, 1.8k for test). However, domain-specific benchmarks often prioritize quality over scale. While smaller in size, they are widely regarded as valuable tools for evaluating specialized capabilities. Several well-regarded examples include:
>
> - VisIT-Bench (592) – Real-world instruction-following tasks [NeurIPS D&B 2023]
> - MM-Vet (218) – Complex visual reasoning [ICML 2024]
> - V*Bench (191) – High-resolution understanding [CVPR 2024]
> - JMMMU handmade subset (600) – Japanese cultural context [NAACL 2025]
>
> These benchmarks, despite their modest scale, are widely used because they offer sufficient discriminative power in specialized domains. Likewise, MangaVQA consists of 526 carefully curated examples. Each question was curated by the seven authors to ensure clarity and avoid subjective interpretation or ambiguous answers. This rigorous process has resulted in a high-quality benchmark that offers consistent and reliable evaluation for the manga domain.
>
> **W2-2: The small size of the Image Understanding category**
>
> Manga storytelling is inherently text-driven, with dialogue, narration, and visual context closely intertwined. Accordingly, our benchmark is designed to promote multimodal-dominant evaluation rather than purely visual or textual analysis. For this reason, we placed the greatest emphasis on the Multimodal Understanding category, which we believe best reflects the core challenge of manga comprehension. In contrast, the Image Understanding category includes fewer questions, as it is particularly difficult to create visual-only questions in manga, where text and images are rarely separable.
>
> **W3: The risk of circular evaluation**
>
> We thank the reviewer for raising an important point. We acknowledge the risk of circular evaluation in the MangaVQA benchmark. To eliminate this risk, we used Gemini 2.5 Flash as an independent judge for evaluation. The results are shown in the tables below. These tables are extended versions of Tables 2 and 3 in the main paper. Even with this alternative judge, the overall trend remains consistent with the original results using GPT-4o as a judge, indicating that our conclusions still hold and the score gain does not come from the circular evaluation. To further address this risk, we use Gemini 2.5 Flash as the evaluation judge for all results presented in this rebuttal, and plan to update the main paper accordingly.
>
> | Method | Judge by GPT-4o (from Table 2)  | Judge by Gemini 2.5 Flash |
> |---|---|---|
> | GPT-4o | 5.76 | 5.88 |
> | Gemini 2.5 Flash | 3.87 | 3.85 |
> | Phi-4-Multimodal | 3.08 | 3.61 |
> | Qwen2.5-VL 7B | 5.36 | 5.51 |
> | MangaLMM | 6.57 | 6.68 |
>
> | FT data | Judge by GPT-4o (from Table 3)  | Judge by Gemini 2.5 Flash |
> |---|---|---|
> | T_OCR | 1.03 | 1.03 |
> | T_VQA | 6.46 | 6.43 |
> | T_OCR + T_VQA (MangaLMM) | 6.57 | 6.68 |
>
> &nbsp;
>
> **W4: Given all text is Japanese, models with strong English but weak JP ability are unfairly penalized.**
>
> Given that manga is inherently Japanese, this setting aligns with real-world usage and appropriately evaluates a model’s ability to handle manga-related tasks in their original linguistic context. We therefore do not consider this evaluation setting to be unfair, but rather necessary to reflect the true demands of the domain.
>
> **W5: Bounding-box noise**
>
> We believe there may be a misunderstanding: the use of bounding-box (bbox) coordinates during fine-tuning is quite different from the setup examined in Table D. In fine-tuning, bbox coordinates are included in the input format of the training OCR dataset (Section 5.1). While the reviewer states that including bbox coordinates “bloats context length without benefit,” we note that this spatial information contributes significantly to OCR performance, as demonstrated in Table 3. In contrast, Table D investigates the effect of including bbox coordinates in the prompts for VQA generation, specifically to assess whether spatial information improves the quality of the generated synthetic questions. We hope this clarification addresses the concern. If not, we would appreciate further feedback from the reviewer.
>
> **W6: Two-page assumption**
>
> This comes from the nature of the manga domain. The reviewer’s statement that “digital readers commonly default to single pages” is not accurate in the context of manga. In Japan, almost all official digital manga platforms default to a two-page spread, not a single page. This reflects the original design of manga, where illustrations and layouts are often created to span across both pages (e.g., Figure 2), making the two-page view essential for proper comprehension.
>
> **W7: one-seed reporting**
>
> To address this concern, we conducted three runs on the main results (Table 3), and the table below shows consistent trends with small standard deviations.
>
> | FT data | MangaOCR  | MangaVQA|
> |---|---|---|
> | T_OCR | 74.5 ± 1.3 | 1.39 ± 0.64 |
> | T_VQA | 0 ± 0 | 6.53 ± 0.12 |
> | T_OCR + T_VQA (MangaLMM) | 71.2 ± 0.6 |  6.70 ± 0.03  |
>
> &nbsp;
>
> **W8: OCR metric rigidity**
>
> As described in Section 6 (Line 212), we adopted standard OCR evaluation metrics commonly used in the field, which consider both the textual content and its position. As detailed in Section A (Line 13), positional information is essential for accurate OCR evaluation in manga, due to its layout-sensitive nature.
>
> Nonetheless, we additionally evaluated models without using positional information. We follow the evaluation metric from TextMonkey, which ignores positional information and checks whether each ground truth string appears somewhere in the predicted text (the “Trans” mode). Even under this setting, models without finetuning performed poorly, reaffirming the value of finetuning on MangaOCR data.
>
> | Method | standard OCR metric | TextMonkey-”Trans” |
> |---|---|---|
> | GPT-4o | 0.0 | 16.1 |
> | Gemini 2.5 Flash | 0.0 | 25.8 |
> | Phi-4-Multimodal | 0.0 | 0.0 |
> | Qwen2.5-VL 7B | 0.9 | 8.1 |
> | MangaLMM | 71.5 | 64.6 |
>
> &nbsp;
>
> **W9: Limited stylistic coverage**
>
> As described in M1 and M2, this study specifically focuses on Japanese comics (manga).
> While we acknowledge that expanding to other comic styles would be interesting, such directions are beyond the scope of this study.

---

### Note · Authors · 2025-08-15

Dear ACs,

Thank you for your time and effort in handling our work. We would like to take this opportunity in the Final Remarks to clarify a point regarding the authors’ rebuttal opportunity.

We would be grateful if you could take into account that authors cannot view Official Comments from reviewers if they are posted after the Author-Reviewer discussion period, and thus cannot respond to them in the Author Final Remarks. Specifically, on 13 Aug (AoE), we received a Mandatory Acknowledgement from Reviewer y1Zw and considered that there might have been an accompanying Official Comment. However, we were unable to confirm its existence or check its content, as comments posted after the discussion period are not visible to authors. If we could have viewed it, we would have responded in our Author Final Remarks.

Finally, we sincerely appreciate the reviews throughout the review process.

Sincerely,

Authors

---

### Decision · Program_Chairs · 2025-09-18

**Decision:**

Reject

**Comment:**

This paper presents a new benchmark for manga OCR and VQA. The motivation is clear, the task design is easy to follow. The authors also release their resources and show that their models outperform existing baselines. However, a major concern is that the benchmark uses the same models both to generate the data and to act as the evaluator, which introduces a risk of evaluation bias. In the rebuttal the authors added experiments with Gemini Flash as the judge, but these are still limited and do not fully resolve the issue of circular evaluation. It also remains unclear whether the models are demonstrating real reasoning ability or mainly learning sequential patterns from the data. Because of this, I think the paper needs modification and another round of review. At this stage, I recommend rejection.